# Automatic classification of prostate MR series type using image content and metadata

**Deepa Krishnaswamy**[1]                                    DKRISHNASWAMY@BWH.HARVARD.EDU
**Bálint Kovács**[2]
**Stefan Denner**[2]
**Steve Pieper**[3]
**David Clunie**[4]
**Christopher P. Bridge**[5]
**Tina Kapur**[1]
**Klaus H. Maier-Hein**[2]
**Andrey Fedorov**[1]

[1] *Brigham and Women's Hospital, Boston, MA, USA*

[2] *Division of Medical Image Computing, German Cancer Research Center, Heidelberg, Germany*

[3] *Isomics, Cambridge, MA, USA*

[4] *PixelMed Publishing, Bangor, PA, USA*

[5] *Massachusetts General Hospital, Boston, MA, USA*

**Editors:** Under Review for MIDL 2024

## Abstract

With the wealth of medical image data, efficient curation is essential. Assigning the sequence type to magnetic resonance images is necessary for scientific studies and artificial intelligence-based analysis. However, incomplete or missing metadata prevents effective automation. We therefore propose a deep-learning method for classification of prostate cancer scanning sequences based on a combination of image data and DICOM metadata. We demonstrate superior results compared to metadata or image data alone, and make our code publicly available at https://github.com/deepakri201/DICOMScanClassification.

**Keywords:** classification, DICOM, MRI, prostate cancer, convolutional neural network

## 1. Introduction

To diagnose clinically significant prostate cancer (PCa), multi-parametric magnetic resonance imaging (MRI) scans are acquired, typically following the PI-RADS v2 guidelines (Weinreb et al., 2016; Turkbey et al., 2019). However, metadata describing the type of scan (free text DICOM field *SeriesDescription* configured by the operator) is prone to user input errors, institutional conventions, and removal during de-identification. As most machine learning (ML) tasks for segmentation and detection of lesions require specific sequences as input (Saha et al., 2023; Adams et al., 2022), efficient curation without relying on free text fields or manual input is necessary. Approaches for series classification have been developed based on metadata (Gauriau et al., 2020; Cluceru et al., 2023), and convolutional neural networks (CNN) using images (Kasmanoff et al., 2023; Salome et al., 2023; van der Voort et al., 2021). However most approaches do not combine metadata with image data in a learned fashion (Cluceru et al., 2023), and relatively few methods have performed classification of prostate/pelvis sequences (Baumgärtner et al., 2023; Helm et al., 2024).

Table 1: Collections from Imaging Data Commons and the corresponding number of MR series (patients in parentheses) included for the analysis. ERC = endorectal coil was used, †=multiple manufacturers, ‡=multiple magnetic field strengths.

| Dataset | With ERC | T2W | DWI | ADC | DCE | Train | Val | Test |
|---|---|---|---|---|---|---|---|---|
| QIN-Prostate-Repeatability (Fedorov et al., 2018) | ✓ | 30 (15) | 30 (15) | 30 (15) | 30 (15) | ✓ | ✓ | ✓ |
| ProstateX† (Litjens et al., 2014, 2017) | – | 431 (346) | 357 (346) | 356 (346) | 15456 (346) | ✓ | ✓ | ✓ |
| Prostate-MRI (Choyke et al., 2016) | ✓ | 26 (26) | 52 (26) | – | 51 (26) | – | – | ✓ |
| Prostate-3T† (Litjens et al., 2016) | – | 64 (64) | – | – | – | – | – | ✓ |
| Prostate-Diagnosis (Bloch et al., 2015) | ✓ | 93 (91) | – | – | – | – | – | ✓ |
| Prostate-MRI-US-Biopsy†‡ (Natarajan et al., 2013) (Sonn et al., 2013) | ✓ | 958 (792) | 110 (108) | 1019 (836) | – | – | – | ✓ |
| Prostate-Fused-MRI-Pathology (Singanamalli et al., 2016) (Madabhushi and Feldman, 2016) | ✓ | 46 (27) | 13 (12) | 12 (12) | 102 (28) | – | – | ✓ |

We propose a CNN-based method for classification of prostate MRI scans, where A) we integrate the image data and DICOM metadata (acquisition parameters captured in fields populated by the scanner) in a single CNN, which has not been done yet for scan classification, B) we train and evaluate our methods using entirely publicly available DICOM PCa MRI collections, and C) we compare our method with a random forest approach using the metadata, and a CNN-based approach using the image data.

## 2. Methodology

**Data** We use publicly available data from NCI Imaging Data Commons (IDC), a cloud-based repository of cancer imaging data (Fedorov et al., 2023) as seen in Table 1. The ground truth series type was assigned semi-automatically by manually checking all possible regular *SeriesDescription* expressions specific for these datasets, where 84.1% of all series were assigned to the T2-weighted (T2W), diffusion-weighted imaging (DWI), apparent diffusion coefficient (ADC) and dynamic contrast enhanced (DCE) classes. Series not included consisted of those acquired sagitally/coronally, localizers, and calculated b-value DWI images. We split the data patient-wise, using 60% of QIN-Prostate-Repeatability (Fedorov et al., 2018) and ProstateX (Litjens et al., 2014, 2017) for training, and 20% of the same datasets for validation. Both collections contain the sequences to classify, including T2W, DWI, ADC and DCE images. We split testing into internal (20% of QIN-Prostate-Repeatability and ProstateX), and external (five collections not seen during training). These external collections were curated in the same manner, to only contain the four classes used for the classification.

**Methods** We propose a CNN-based method that leverages image data and DICOM metadata. The following metadata attributes were used, as these are machine-generated, standardized, and not removed during de-identification: *RepetitionTime*, *EchoTime*, *FlipAngle*, *ScanningSequence*, and *ContrastBolusAgent*. A derived *is4D* attribute was assigned based on whether spatially overlapping slices were detected within the series. Feature scaling was applied to relevant metadata. Figure 1 summarizes the metadata distribution. Center slices were extracted from 3D volumes, resampled to 64x64, and normalized between 0 and 1. The CNN consisted of three convolution layers each followed by max pooling, followed by three

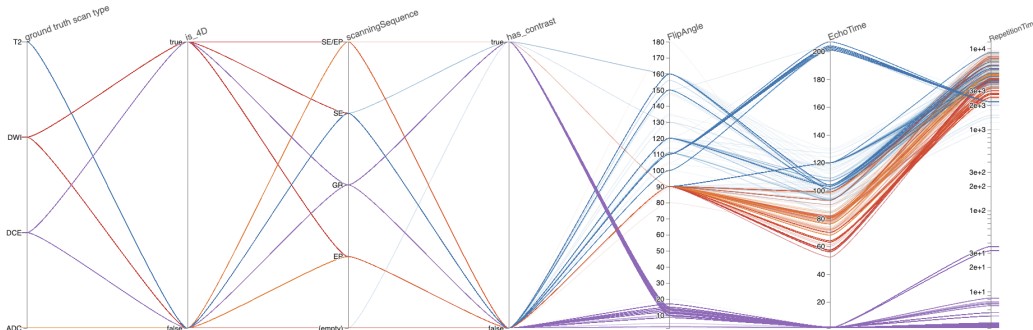

Figure 1: Hiplot visualization (Haziza et al., 2020) of DICOM metadata parameters, colored by the assigned ground truth scan type.

Table 2: Quantitative results for the three methods on the two test datasets. The mean is provided for the F-beta score when four fold cross-validation is performed.

| Method | Internal T2Ax | Internal DWI | Internal ADC | Internal DCE | External T2Ax | External DWI | External ADC | External DCE |
|---|---|---|---|---|---|---|---|---|
| Metadata | 1.00 | 1.00 | 1.00 | 1.00 | 0.98 | 0.60 | 0.91 | 1.00 |
| Images | 0.99 | 0.99 | 0.99 | 0.99 | 0.93 | 0.59 | 0.99 | 0.89 |
| Images + metadata | 1.00 | 1.00 | 0.99 | 0.99 | 0.98 | 0.72 | 0.99 | 0.99 |

dense layers, where the metadata was concatenated. Sparse categorical cross entropy loss was used with Adam optimization. K-fold cross validation was performed (4 folds), where each model was trained for 10 epochs with early stopping, and probability outputs were ensembled. The combined image and metadata approach was compared to A) a random forest classifier using only the metadata, and a B) CNN model utilizing only the images.

## 3. Results and Discussion

Table 2 displays the evaluation. We note the higher accuracy on the internal dataset, as patients are from the same collection as training/validation, compared to the external test set, which contains five collections not seen during training. The metadata-only approach performs poorly on the external test set due to DWI misclassified as ADC and vice versa. Depending on the format of the DWI data (single series is a 4D volume, or multiple series each with a 3D volume), the latter can be considered similar to ADC. In the images and metadata combination approach, the performance of DWI in the external test improves, but DWI still suffers from misclassification as ADC due to the *is4D* parameter, and the intensity similarity of the two. Future work involves refining the metadata used, and including the ability to differentiate between low and high b-value images.

## Acknowledgments

We acknowledge Dr. Clare Tempany, grants P41EB028741-04, P41EB028741-03S1, and NIH NCI under Task Order No. HHSN2611 0071 under Contract No. HHSN261201500003l, which made this research possible.

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
