# OpenReview forum: "Automatic classification of prostate MR series type using image content and metadata"
_MIDL.io/2024/Short_Papers — MIDL 2024 Short Papers_

### Official Review · Reviewer_2Y3c · 2024-04-24

**Confidence:** 4
**Final Rating:** 3.5

**Review:**

This paper proposes a method for classifying type of MR scan from MR prostate images and DICOM metadata using CNNs.

Pros:
- Reproducibility is high, as the experiments use all public datasets from multiple different studies/sites, and code the is also released.
- Validation is performed on both in-domain and out-of-domain data
- The paper is clearly written with good detail of methods, settings
- The problem of classifying unknown sequence types is important for making the most of available medical imaging data

Cons:
- Technical innovation is limited as method applies standard method of CNN + MLP fusion of image + meta data
- Impact of approach is not clear to me - the meta data likely does better than image or image+metadata for in domain experiments and does as good or better for a few of the sequences for out of domain. Also, for the out of domain experiments only 2 of the datasets were used for training. If more of the available data were used for training and then model tested for example in a leave one dataset out framework, I am wondering of the metadata may largely outperform the image + metadata approach, similar to in domain.

---

### Decision · Program_Chairs · 2024-04-26

Accept